# The Association between Sleep and Chronic Spinal Pain: A Systematic Review from the Last Decade

**DOI:** 10.3390/jcm10173836

**Published:** 2021-08-26

**Authors:** Eveline Van Looveren, Thomas Bilterys, Wouter Munneke, Barbara Cagnie, Kelly Ickmans, Olivier Mairesse, Anneleen Malfliet, Liesbet De Baets, Jo Nijs, Dorien Goubert, Lieven Danneels, Maarten Moens, Mira Meeus

**Affiliations:** 1Spine, Head and Pain Research Unit Ghent, Department of Rehabilitation Sciences and Physiotherapy, Faculty of Medicine and Health Sciences, Ghent University, 9000 Ghent, Belgium; thomas.bilterys@vub.be (T.B.); barbara.cagnie@ugent.be (B.C.); lieven.danneels@ugent.be (L.D.); mira.meeus@ugent.be (M.M.); 2Pain in Motion Research Group (PAIN), Department of Physiotherapy, Human Physiology and Anatomy, Faculty of Physical Education and Physiotherapy, Vrije Universiteit Brussel, 1090 Brussels, Belgium; wouter.munneke@vub.be (W.M.); kelly.ickmans@vub.be (K.I.); anneleen.malfliet@vub.be (A.M.); liesbet.de.baets@vub.be (L.D.B.); jo.nijs@vub.be (J.N.); 3Pain in Motion International Research Group, 1090 Brussels, Belgium; goubertdorien@gmail.com; 4Research Foundation—Flanders (FWO), 1000 Brussels, Belgium; 5Sleep Laboratory and Unit for Chronobiology U78, Brugmann University Hospital, Université Libre de Bruxelles—Vrije Universiteit Brussel, 1020 Brussels, Belgium; olivier.mairesse@vub.be; 6Brain, Body and Cognition, Department of Psychology, Faculty of Psychology and Educational Sciences, Vrije Universiteit Brussel, 1050 Brussels, Belgium; 7Department of Physical Medicine and Physiotherapy, University Hospital Brussels, 1090 Brussels, Belgium; 8Unit of Physiotherapy, Department of Health and Rehabilitation, Institute of Neuroscience and Physiology, Sahlgrenska Academy, University of Gothenburg, 405 30 Gothenburg, Sweden; 9University of Gothenburg Center for Person-Centred Care (GPCC), Sahlgrenska Academy, University of Gothenburg, 405 30 Gothenburg, Sweden; 10Department of Neurosurgery and Radiology, University Hospital, 1090 Brussels, Belgium; maarten.TA.moens@vub.be; 11Department of Manual Therapy, Faculty of Medicine and Pharmacy, Vrije Universiteit Brussel, 1090 Brussels, Belgium; 12Department of Rehabilitation Sciences and Physiotherapy (MOVANT), Faculty of Medicine and Health Sciences, University of Antwerp, 2610 Antwerp, Belgium

**Keywords:** chronic low back pain, chronic neck pain, sleep, insomnia, systematic review

## Abstract

Chronic spinal pain, including both neck and low back pain, is a common disabling disorder in which sleep problems are frequently reported as a comorbidity. The complex processes of both sleep and chronic pain seem to have overlapping mechanisms, which may explain their often established bidirectional relationship. This systematic review aims to investigate the assumed association between sleep and chronic spinal pain by providing an overview of the literature from the last decade. Eligible studies were obtained by searching four databases (PubMed, Embase, Web of Science, and PsycARTICLES). Articles were found relevant if they included a human adult population and investigated the possible association between sleep parameters and chronic spinal pain. Only studies published after January 2009 were included, as this review aimed to provide an update of a previous literature overview on this topic. The quality of the studies was assessed by risk of bias and level of evidence. A total of twenty-seven studies (6 cohort, 5 case-control, and 16 cross-sectional studies) were included in this systematic review. The methodological quality of these studies was low to moderate. The majority of studies reported weak to moderate evidence for an association between sleep parameters and chronic spinal pain, with more severe pain accompanied by more disturbed sleep. Addressing frequently reported sleep problems in chronic spinal pain patients therefore appears to be a necessary complement to pain management to achieve optimal treatment outcomes.

## 1. Introduction

Spinal pain consisting of both neck and low back pain is an ubiquitous disorder whereby a substantial number of patients develop recurrent or chronic complaints [1,2,3]. The underlying pathophysiology of the chronic variant points in the direction of alterations in the central nervous system and also involves psychological factors [4,5,6]. Besides a major socio-economic impact on both the patient and society, the disabling character of chronic spinal pain (CSP) also negatively affects quality of life parameters and sleep [7]. Sleep disturbances, which impair the multifunctional protective role of sleep on physiological homeostasis and restoration [8], are frequently reported by spinal pain patients, with a prevalence up to more than 50 percent [9,10,11,12,13].

There is growing evidence that the neurobiology of chronic pain shows overlapping mechanisms with sleep problems, which may explain the often established bidirectional relationship between chronic pain and sleep problems in people with fibromyalgia, osteoarthritis, and chronic low back pain (CLBP) among others [14,15,16,17,18]. Longitudinal and experimental research has affirmed that disturbed sleep causes generalized hyperalgesia and reduced endogenous pain inhibition in healthy subjects [19,20]. Increased pain can, in turn, further disrupt sleep, leading to a vicious cycle that can be further exacerbated by pain medication adversely affecting sleep [21,22]. However, insomnia is increasingly assumed to be a primary determinant, as it both deteriorates existing pain and predicts pain onset [14]. Insomnia disorder is defined as the patient’s dissatisfaction with the quality or quantity of his or her sleep, together with other symptoms manifesting during the day or night such as fatigue, mood disturbances, and cognitive complaints over the course of at least three nights a week for a period exceeding three months [23].

The link between sleep and pain has already been clinically studied in specific pain populations, suggesting disorder-specific disturbances in sleep architecture [14,15,16,17,18]. In the past decade, substantial research effort has focused on the relationship between sleep and chronic pain [14,15,24]. Specific for the association between sleep and CLBP, a previous systematic review showed considerable evidence that CLBP is related to sleep outcomes, with more pain accompanied by more sleep disturbances [16]. Sleep parameters that are affected in this patient population compared to controls were sleep quantity, sleep disturbance, sleep quality, next day functioning, and sleep onset latency [16]. Since the database search in the aforementioned review was performed up to January 2009, the findings are rather outdated. Indeed, research into the association between sleep and spinal pain is evolving rapidly, and the literature in this domain has greatly increased over the past decade. In addition, to date, there is no review that has been conducted on the association between sleep and chronic neck pain (CNP). However, due to the similar underlying physiological and psychological pain mechanisms [25], a similar link with sleep as seen in CLBP could be expected in patients reporting CNP. Therefore, the aim of the current systematic review is to provide an update of the existing review of Kelly et al. (2011) regarding the relationship between CLBP and sleep [16]. However, the research question was specified to the presence of a bidirectional association between sleep parameters (i.e., sleep quality, sleep duration, insomnia severity, and sleepiness) and CSP parameters (i.e., pain duration, pain intensity, pain sensitivity) in both cross-sectional and longitudinal studies. Consequently, this included exploring the association between the onset of CSP as a result of insomnia. In addition, the pain outcome was extended to CSP, including chronic low back or neck complaints, or both.

## 2. Methods

This systematic review was conducted according to the Preferred Reporting Items for Systematic Reviews and Meta-Analyses (PRISMA) guidelines [26] and was listed in PROSPERO with the registration number CRD42021233659 (https://www.crd.york.ac.uk/prospero/display_record.php?ID=CRD42021233659, accessed on 6 April 2021).

### 2.1. Information Sources and Search Strategy

A total of four electronic databases, i.e., PubMed (https://pubmed.ncbi.nlm.nih.gov, accessed on), Embase (http://embase.com, accessed on 7 July 2020), PsycARTICLES (https://search.proquest.com/psycarticles, accessed on 7 July 2020), and Web of Science (http://webofscience.com, accessed on 7 July 2020), were searched on 7 July 2020 for relevant articles to be included in this systematic review. The PECO(S) (patient, exposure, comparison, outcome, and study design) approach was used to define the subsequent search requests: (1) “What is the association between sleep properties (E/O) and chronic spinal pain parameters (E/O)?”; (2) “How does chronic spinal pain (E) predict sleep disturbance (O)?”; (3) “How does sleep disturbance (E) predict chronic spinal pain (O)?”. Thus, a variable population consisting of subjects with CSP and/or insomnia as well as baseline healthy individuals was included and the occurrence of CSP and/or insomnia over time was examined. Because this review specifically investigated the association between sleep parameters and CSP, an alternation of these components as outcome (O) and exposure (E) possibly occurs throughout different studies.

Validated questionnaires or sleep diaries are often applied to assess patient-reported sleep parameters such as insomnia severity [27], sleepiness [28], and sleep quality [29,30] in both research activities and clinical practice. Via polysomnography and actigraphy, an objective sleep indication can be obtained that provides insight into sleep quantity parameters such as total sleep time (TST), sleep onset latency (SOL), awakenings after sleep onset (WASO), and sleep efficiency (SE). In addition, polysomnographic results reflect the different sleep stages, ranging from slow wave sleep to rapid eye movement (REM) sleep [31].

The assessed pain outcomes in this review are pain intensity and duration in CSP patients and the occurrence of CSP in a health survey population, which were all measured with questionnaires. Furthermore, quantitative sensory testing is a frequently used psychophysical method to measure pain sensitivity as the outcome parameter [32].

The search strategy, which was formulated by one author (E.V.L.) and controlled by a second experienced investigator (M.M.), was built by combining key words and MeSH terms relating to “sleep” and “spinal pain” with the appropriate Boolean operators (Table 1 and Appendix A).

### 2.2. Eligibility Criteria and Study Selection

Studies were eligible for inclusion if following criteria were fulfilled: (1) written in English or Dutch; (2) population consisting of human adults (≥18 years); (3) published after January 2009 since this was the final date of the search in the study of Kelly et al. (2011) [16]; (4) presence or onset of CSP defined as low back or neck pain that persisted for ≥12 weeks; and (5) studies investigating the association between pain (i.e., pain intensity, pain duration, pain sensitivity) and sleep (sleep quality, sleep duration, insomnia severity, sleepiness). There were two independent researchers (E.V.L. and T.B.), both holding a master’s degree in physiotherapy and rehabilitation sciences, who screened the obtained articles based on title and abstract in a first phase. Subsequently, in a second phase, eligible studies were selected based on the full text. In case of disagreement, a discussion between the two researchers was held until a consensus was reached.

### 2.3. Risk of Bias in Individual Studies

The risk of bias of the incorporated studies was independently assessed by two researchers (E.V.L. and W.M.) to determine the methodological quality. Results of the individual risk of bias analyses were compared, and irregularities in the scores were discussed in order to reach a consensus. Depending on the design of the studies, specific versions of the Newcastle-Ottawa Scale (NOS) were applied, originating from the website of the Dutch Cochrane Center (https://netherlands.cochrane.org, accessed on 12 August 2020). The recommended NOS for cohort and case-control studies was used by applying a rating system to score the subcategories of selection, comparability, and exposure or outcome (http://www.ohri.ca/programs/clinical_epidemiology/oxford.asp, accessed on 12 August 2020). An adapted version of the NOS was used to score the cross-sectional studies. The legends of Table 2, Table 3 and Table 4 provide more details on the various topics that were assessed.

The selection criteria were similarly interpreted across the different study designs, whereby a representative sample was required to be recruited through various channels and over a sufficiently extensive region in order to obtain a positive score. If the population included had CSP and/or sleep problems, the ascertainment of exposure or case definition had to be based on the findings of quantitative sensory testing or magnetic resonance imaging or an objective sleep analysis, respectively. Furthermore, a positive score was assigned in the cross-sectional studies if a power calculation or rule of thumb was applied to justify the sample size. On the one hand, the comparability subcategory was scored positively if the study corrected for age, and sex and on the other hand, if psychological factors (i.e., anxiety and/or depression) were controlled [58]. Regarding the exposure or outcome subcategory, as the primary research question investigated the association between sleep and CSP, these parameters could be interpreted both as outcome and exposure within the included studies. Specifically, for pain, the results of a quantitative sensory test or magnetic resonance imaging were needed to positively score the relevant item. Additionally, if sleep disturbance was either the outcome or exposure, an objective sleep measurement (i.e., polysomnography) was required to satisfy the item in question. In the cohort studies, a follow-up period of 3 months was considered relevant in concerning the sleep and pain association [23,59]. A loss-to-follow-up rate of 20 percent was assumed to be relevant, as this is generally considered acceptable in cohort studies [60].

After assessing the risk of bias, the level of evidence was assigned to each individual study based on the study design and methodological quality according to the 2005 classification system of the Dutch Institute for Healthcare Improvement (CBO) (Table 5). Finally, clusters of studies were created, in which studies with homogenous procedures and pain populations (neck and/or low back pain) were grouped to determine the strength of conclusion (SoC) for each cluster while considering the design and the risk of bias of the studies (Table 6).

### 2.4. Data Collection Process

Relevant information regarding the association between sleep and chronic spinal pain was extracted from eligible articles and was summarized in an evidence table (Table 7) representing the following items: first author, study design, sample characteristics, outcome measures, associations between sleep and pain, and main results with corresponding statistical values. Pertinent data were collected in the evidence table by the first author (E.V.L.) and was checked by T.B., who acted as a second reviewer.

## 3. Results

### 3.1. Study Selection

Searching the databases for relevant studies yielded 3764 articles. After removing duplicates, the remaining 2802 studies were screened by title and abstract and subsequently by full text, resulting in 27 eligible studies that were included in the qualitative synthesis. The selection process of these articles is presented in Figure 1.

### 3.2. Risk of Bias and Level of Evidence

The results of the risk of bias analysis are shown in Table 2, Table 3 and Table 4 for cohort, case-control, and cross-sectional studies, respectively. Assessing the risk of bias resulted in a similar score between the two raters for the majority of the items (204/235; 86.8%). After discussing the items without agreement, a consensus was reached on the final score.

Low scores for the selection criteria were mainly found in cohort and cross-sectional studies. This was primarily due to a lack of pain data obtained via quantitative sensory testing, magnetic resonance imaging, or sleep information based on polysomnographic data. In addition, the other selection criteria (representativeness of sample, sample size, and non-respondents) frequently received a negative score in the cross-sectional studies. Comparability regarding age and sex was present in almost all studies. However, only slightly over half of the studies also controlled for psychological factors. The results of the subcategory of exposure or outcome showed a high risk of bias. Again, the method of the outcome assessment is the main issue in the negative scoring. Moreover, in case-control studies, information on the non-response rate is missing, which negatively affects the score.

All of the cohort and case-control studies received a level of evidence of B (Table 2 and Table 3). The cross-sectional studies were given a level of evidence of C (Table 4).

### 3.3. Study Characteristics

Of the 27 included studies, six were prospective cohort studies [33,34,35,36,37,38], five were case-control studies [39,40,41,42,43], and the remaining sixteen studies had a cross-sectional study design [11,12,13,44,45,46,47,48,49,50,51,53,54,55,56,57]. The case-control studies [39,40,41,42,43] compared CLBP or patients with whiplash associated disorders with pain-free healthy individuals.

The sample size of the selected articles varied from 30 [42] to 30,699 [46]. All of the studies covered both men and women. Only one study reported results of a mixed population including both chronic low back and neck pain patients [11]. More detailed study characteristics are presented in the evidence table (Table 7).

The results obtained from the 27 included studies are subdivided by the various outcome parameters. The first part of the results summarizes the associations between sleep and pain examined in predominantly cross-sectional and case-control studies. A second section elaborates on the causality of the relationship between sleep and CSP based on cohort research. Multiple questionnaires and corresponding interpretations of the scores obtained were applied throughout the studies. Table 8 provides an overview of the used assessment tools and related details.

### 3.4. Synthesis of Results

A summary of the main results of the included studies is presented in the evidence table (Table 7).

### 3.5. Associations between Sleep Parameters and Chronic Spinal Pain

#### 3.5.1. Sleep Quality

##### Association between Sleep Quality and Pain Intensity

The correlation between sleep quality and pain intensity in patients with CLBP or CNP was examined in six studies of which three were case-control studies [40,42,43] and three had a cross-sectional design [45,47,52]. A total of two of these studies investigated the association between the prior night’s sleep quality and both subsequent [40,45] and hour by hour pain intensity in more detail [45]. Quality of sleep was assessed using the Pittsburgh Sleep Quality Index (PSQI) [40,42,43,47,52] or a single self-reported questionnaire [45]. Pain intensity was examined by the McGill Pain Questionnaire (MPQ) [40,43,52], the Numeric Pain Rating Scale (NRPS) [52], a Visual Analogue Scale (VAS) [47], the Bodily Pain Scale (PBS) of the SF-36 [42], or a self-reported questionnaire [45].

All of the studies except one reported a significant negative correlation between sleep quality and pain intensity in patients with CLBP [40,42,43,45,52] or CNP [47]. However, in contrast to the Spanish participants with CLBP who reported worse sleep quality in the cross-sectional study of Rodrigues-De-Souza et al. (2016), no significant correlation between sleep quality and pain intensity was found in the Brazilian subjects reporting CLBP [52].

A significant negative association was found between the prior night’s sleep quality and subsequent pain intensity in the morning and not in the afternoon or evening, indicating that higher pain levels are only present in the morning after a poor night of sleep [40,45]. Conversely, the association between the prior day’s pain intensity and the subsequent night’s sleep quality was not significant [45].

##### Association between Sleep Quality and Pain Duration

A total of two case-control studies [41,43] and one cross-sectional study [54] investigated the association between sleep quality and pain duration in a CLBP population. The PSQI that was used in all three studies to assess the quality of sleep and the duration of back pain was self-reported by the subjects [41,43,54].

No significant correlation between sleep quality and pain duration was found in the study of Hong et al. (2014) [41]. Similarly, no significant association between sleep quality and pain duration (>1 year vs. ≤1 year [54]; >11 years vs. ≤11 years [43]) was reported, although a significantly worse daytime dysfunction sub-score of the PSQI was found if the pain duration was more than eleven years [43].

##### Association between Sleep Quality and the Presence of CLBP

There was one case-control study that investigated the association between sleep quality and CLBP [43]. Sleep quality was assessed by the Pittsburgh Sleep Quality Index (PSQI) [43]. Mechanic low back pain was diagnosed by a physician [43].

Significantly higher odds of poor sleep quality were found in subjects reporting CLBP compared to subjects without CLBP [43].

##### Association between Sleep Sufficiency and CLBP

There was one cross-sectional study that investigated the association between sleep sufficiency and CLBP [57]. Both parameters were assessed by self-reported questionnaires [57]. Significantly higher odds of CLBP were indicated in patients with poorer sleep sufficiency [57].

#### 3.5.2. Insomnia and Sleep Disturbances

##### Association between Insomnia Severity and Pain Intensity

The association between insomnia and pain intensity in patients with CLBP [12,13,42,49,55,56] or CNP [50] was investigated in six cross-sectional studies and one case-control study. Of these, six of these studies specifically examined the correlation between insomnia severity and pain intensity in a population with CLBP [12,13,42,49,55,56]. In addition, three studies assessed the association between these parameters in terms of odds ratios in patients with CLBP [49,56] or CNP [50]. The presence and severity of insomnia was assessed by the Insomnia Severity Index (ISI) [12,13,42,49,50,56] and the Athens Insomnia Scale (AIS) [55]. Back or neck pain intensity was assessed by a Visual Analogue Scale (VAS) [12,55,56], a Numeric Rating Scale (NRS) [13,49,50], or the Bodily Pain Scale of the SF-36 version 2 [42].

All six studies that investigated the correlation between insomnia severity and pain intensity reported a significant positive correlation coefficient [12,13,42,49,55,56]. Furthermore, higher odds of insomnia were found in patients with severe CLBP [49,56] and CNP [50] compared to low pain severity (VAS/NRS < 7).

##### Association between Sleep Disturbance and Pain Intensity

The association between sleep disturbance and pain intensity in patients with CLBP was investigated in one cross-sectional study using the short forms of the PROMIS sleep disturbance and the McGill Pain Questionnaire respectively [44].

A significant positive correlation was found between the pain ratings and sleep disturbance, indicating that higher pain intensity, sensory, and affective pain ratings are associated with more severe sleep disturbance in CLBP patients [44].

##### Association between Insomnia Severity and Pain Duration

There were four cross-sectional studies that investigated the correlation between the severity of insomnia, assessed by the Insomnia Severity Index (ISI) [12,49,50] or the Athens Insomnia Index (AIS) [55], and the self-reported duration of pain in patients with CLBP or CNP.

No significant correlation was found between insomnia severity and pain duration in two studies regarding CLBP [12,49] and one study on CNP [50]. Uchmanowicz et al. (2019) found insomnia severity to be negatively correlated with CLBP duration, indicating that more severe insomnia is associated with a shorter duration of pain symptoms (>3 months) [55].

##### Association between Insomnia Severity and CLBP

There was one cross-sectional study that investigated the association of insomnia and CLBP [46]. Both parameters were assessed by self-reported questionnaires.

Significantly higher odds of CLBP were found in patients with insomnia compared to patients without insomnia [46].

##### Association between Sleep Disturbance and CLBP

There was one cross-sectional study [53] and one case-control study [41] that assessed the odds ratios representing the presence of sleep disturbance, which was assessed with the Pittsburgh Sleep Quality index (score > 5) or a self-reported questionnaire, in patients with CLBP.

The cross-sectional study found higher odds of sleep disturbance in CLBP patients compared to subjects without CLBP [53]. However, the case-control study found no significant correlation between the aforementioned parameters [41].

##### Association between Sleep Problems and CNP

The association of sleep problems in patients with chronic whiplash associated disorder was assessed in one case-control study [39]. The sleep problems were self-reported based on a single questionnaire.

Higher odds of sleep problems were found in patients with whiplash associated disorder compared to healthy control subjects [39].

#### 3.5.3. Sleep Duration and Sleep Sufficiency

##### Association between Sleep Deprivation and Pain Intensity

There was one cross-sectional study that investigated the correlation between the grade of sleep deprivation, whereby the sleep data were selected from the Oswestry Disability Questionnaire (ODQ) and pain intensity, based on the score of the Numeric Rating Scale (NRS) in a mixed population of both CLBP and CNP patients [11].

A positive correlation between sleep deprivation and pain intensity was found, indicating that a higher degree of sleep disturbance is associated with higher pain intensity [11]. In addition, a significant difference between pain intensity subgroups (NRS < 5 vs. ≥ 5; NRS < 5 vs. > 7) was found, with a higher pain intensity associated with a higher degree of sleep deprivation [11].

##### Association between Objective Sleep Parameters and Pain Intensity

Only one case-control study investigated the correlation between objective sleep parameters (total sleep time, sleep latency, sleep efficiency, and awake time after sleep onset) measured with actigraphy and pain intensity assessed with the Bodily Pain Scale of the SF36-v2 in CLBP patients [42].

This study reported a strong correlation between the objective sleep parameters and pain intensity, suggesting that poorer sleep was associated with more severe pain in patients with CLBP [42].

##### Association between Sleep Duration and CLBP

The association between sleep duration and CLBP, assessed with self-reported questionnaires and the Japanese Orthopedic Association Back Pain Evaluation Questionnaire (JOABPEQ) [51], was investigated in two cross-sectional study [51,57].

There was one study that reported significantly higher odds of CLBP in people with a shorter sleep duration (<5 h vs. ≥6 to <7 h; ≥5 h to <6 h vs. ≥6 to <7 h) [57]. In addition, a significant decrease in odds of CLBP was found for a sleep duration of 7 to 8 h per night compared to the reference sleep duration of 6 to 7 h [57]. However, a sleep duration of more than 8 h per night did not result in a significant association with CLBP [57]. Ouchi et al. (2019) found higher odds of a sleep duration of less than 6 h in CLBP patients compared to a sleep duration of 6 to 7 h per night [51]. No association was present in this study for a sleep duration of more than 7 h per night and CLBP compared to 6 to 7 h [51].

#### 3.5.4. Sleepiness

#### Association between Sleepiness and Pain Intensity

There was one cross-sectional study that investigated the correlation between sleepiness and pain intensity in patients with CLBP [55]. Probability of falling asleep during daily life activities was assessed with the Epworth Sleepiness Scale (ESS), while the currently experienced pain intensity was investigated with a Visual Analogue Scale [55].

A significant positive correlation was found between sleepiness and pain intensity in CLBP patients, indicating higher sleepiness being associated with higher pain intensity [55].

#### Association between Sleepiness and Pain Duration

The correlation between sleepiness, assessed with the Epworth Sleepiness Scale (ESS), and self-reported pain duration in CBLP patients was investigated in one cross-sectional study [55].

A significant negative correlation was found between these parameters, which that implies higher sleepiness is associated with shorter pain duration (>3 months) [55].

### 3.5.5. Summary of Results for the Association between Sleep Parameters and CSP

In summary, weak to moderate evidence was found for a correlation between sleep parameters and pain intensity, indicating that higher pain intensity levels are correlated with poorer sleep quality (SoC 2 in CLBP; SoC 3 in CNP), more severe insomnia symptoms (SoC 2 in CLBP; SoC 3 in CNP), more severe sleep disturbance (SoC 3 in CLBP), shorter sleep duration (SoC 3 in CSP), worse objective actigraphy-derived sleep parameters (SoC 3 in CLBP), and higher sleepiness (SoC 3 in CLBP) in patients reporting CSP. In addition, particularly in individuals with CLBP, weak to moderate evidence was found for the following associations with sleep parameters: higher odds of CLBP were obtained in patients with more severe insomnia symptoms, shorter sleep duration, and poorer sleep sufficiency (SoC 3). Moreover, higher odds were found for poor sleep quality and sleep disturbances in patients with CLBP compared to healthy controls (SoC 3). Proof for an association between CNP and sleep is limited to weak evidence for higher odds of sleep problems in patients with chronic whiplash associated disorder (SoC 3).

Weak to moderate evidence was found for the absence of a correlation between sleep quality (SoC 2 in CLBP) or insomnia severity (SoC 3 in CLBP; SoC 3 in CNP) and pain duration. However, contrary to the aforementioned findings, weak evidence was present for a negative correlation between pain duration and severity of insomnia or sleepiness in CLBP patients (SoC 3).

A brief overview of these results is provided in Table 9.

## 3.6. Sleep Problems as Predictor for CSP

In six prospective cohort studies, the association between sleep problems on pain was investigated [33,34,35,36,37,38]. Of these studies, four of them examined the association between insomnia or sleep problems with the development of C(L)BP [33,35,38] or CNP [34,35,38] at a second (or third) point in time based on odds [33,34] and risk ratios [35,38]. There was one study that examined the association between sleep problems more specifically with pain intensity [36]. In addition, two studies investigated the association between sleep problems and recovery from CLBP [36,37]. In more detail, one study examined the impact of sleeplessness and number of insomnia symptoms in relation to recovery in male and female subjects [37]. Furthermore, the association between the type of sleep problems (developing, persistent or resolving) and recovery from CLBP in terms of pain intensity was investigated [36]. The sleep parameters in these studies were examined by means of a self-reported questionnaire [35,38], the Athens Insomnia Scale (AIS) [33], the Jenkins Sleep Questionnaire [34], or the Pittsburgh Sleep Quality Index (PSQI) [36]. Self-reported questionnaires [34,35,38] or a medical interview [33] were used to assess CLBP or CNP. Pain intensity was measured with a Visual Analogue Scale (VAS) [36].

Agmon et al. (2014) found higher odds of CBP in patients with insomnia compared to subjects without insomnia during a prospective cohort study [33]. Both the studies of Mork et al. (2014) and Uhlig et al. (2018) showed a higher risk for CLBP and CNP at follow-up relative to baseline sleep problems [35,38]. More specifically, patients who reported often or always having sleep problems had a higher risk of developing CLBP or CNP [35]. In addition, Kääria et al. (2012) found higher odds of CNP in women with rare to occasional sleep problems. This association was even stronger in both men and women reporting sleep problems frequently, resulting in higher odds of CNP [34]. A similar result was found in the study of Pakpour et al. (2018), whereby higher odds of CLBP intensity were reported in patients with baseline sleep disturbances, meaning that higher pain levels were present in patients with poor sleep at the start of the study [36]. Furthermore, higher odds of non-recovery in terms of pain intensity (VAS ≤ 1) in CLBP patients who develop sleep problems or who have persistent sleep problems regarding baseline were found in this study [36]. In contrast, lower odds of non-recovery in terms of pain intensity were found in patients with resolving sleep problems, indicating that as sleep problems diminish over time, CLBP will also improve [36]. Skarpsno et al. (2019) showed a higher risk of non-recovery from CLBP in both men and women if they reported sleeplessness often or always [37]. In addition, a higher risk of non-recovery from CLBP was found in women reporting a higher number of insomnia symptoms. No association between the number of insomnia symptoms and non-recovery from CLBP was found in male subjects [37].

### Summary of Results for Sleep Problems as Predictor for CSP

Regarding the role of sleep disturbance as a predictor for CSP, moderate evidence was found for a negative impact of sleep problems on the occurrence of CSP (SoC 2). Weak evidence was present for higher CLBP pain levels at follow-up in patients reporting having problems with sleep (SoC 3). In addition, regarding recovery from CLBP, weak evidence was found for higher odds of non-recovery in people who develop or have persistent sleep problems over time or who report a high degree of sleeplessness (SoC 3). Additionally, weak evidence indicated that a higher number of insomnia symptoms was predictive for non-recovery from CLBP in women but not in men (SoC 3). The found associations are presented in Table 10.

## 3.7. CSP as Predictor for Sleep Problems

There was one prospective cohort study that investigated the association between pain on sleep [33]. This study examined the association between changes in CLBP (time point 1 to time point 2) reported during an interview and changes in insomnia (time point 2 to time point 3) assessed by the Athens Insomnia Severity (AIS) [33].

No significant associations were found between changes in CLBP in the function of change in insomnia [33].

### Summary of Results for CSP as Predictor for Sleep Problems

Finally, weak evidence was found that CLBP was not a predictor for insomnia (SoC 3). This result is also shown in Table 10.

## 4. Discussion

The purpose of this systematic review was to provide an overview of the evidence from the last decade regarding the association between sleep and chronic spinal pain. To comply with this research question, eligible studies on the interaction between sleep and chronic spinal pain variables were found. The data extraction of the 27 included studies resulted in a majority reporting the presence of an association between sleep and chronic spinal pain parameters [11,12,13,34,35,36,37,38,39,40,41,43,45,46,47,49,50,51,52,53,54,55,56,58,59,60]. Thus, sleep quality, insomnia and sleep deprivation severity (i.e., sleep quantity), sleep disturbance, and sleepiness were found to be correlated with pain intensity in such a way that higher pain intensity levels were found in CSP patients reporting poorer sleep. However, no significant association was found for these sleep parameters and pain duration, except the finding that higher sleepiness was correlated with shorter pain duration in CLBP patients. These results confirm the previously reported findings in the review of Kelly et al. (2011), which point to an association between sleep and pain in patients with CLBP [16]. More specifically, the review reported CLBP to be associated with greater sleep disturbances and a shorter sleep duration [16]. To these findings, the current review adds the presence of a negative association between CNP and the sleep parameters studied and presents the role of sleep problems as a predictor for CSP or for the non-recovery of CLBP but not vice versa.

The etiology of chronic pain has a crucial role with respect to the underlying neurobiological processes regarding the association between sleep and pain. As such, whether the pain has a central or peripheral, an inflammatory, or structural origin, will affect its relationship with sleep [61,62]. Various mechanisms to explain the association between sleep and chronic pain have been suggested. Both experimental and longitudinal studies have indicated that modifications in the facilitation and inhibition of pain processes may constitute overlapping mechanisms in the relationship between sleep and chronic pain. Accordingly, sleep disturbances affect the immune system, causing a shift towards a presumably glia-mediated pro-inflammatory state, resulting in hyperalgesia [25,63,64]. Psychological factors such as chronic stress, mood, and anxiety also contribute to the increased pain sensitivity via the mediating role of the hypothalamus–pituitary–adrenal (HPA) axis [25,63]. Additionally, dysfunctional monoaminergic pathways and changes in endogenous substances (orexin, vitamin D, nitric oxide) seem to be involved in the alterations in endogenous pain modulation due to sleep disturbances [25,63]. Moreover, interactions also occur between these various systems and mediators. For example, it is suggested that pain sensitivity follows a physiological rhythmicity modulated by both the sleep-independent circadian rhythm and homeostatic sleep pressure [65]. Therefore, the evidence indicates the existence of an endogenous circadian rhythm in the processing of afferent sensory information at the dorsal horn of the spinal cord. Specifically for nociceptive C fibers accounting for slow-conducting diffuse pain sensations and involved in chronic pain, a peak in pain sensitivity was found during the night between 12 a.m. and 3 a.m. [65]. In addition, top-down pain inhibition in the dorsal horn seems to follow a rhythmicity, but it may not be affected by the circadian process. Instead, building up sleep drive seems to affect the top-down pain inhibition, resulting in an increase in pain sensitivity, especially in chronic pain [65]. Taken together, poor sleep results in an exacerbation in responses to nociceptive stimuli and thus increases in existing pain hypersensitivity with variations in pain sensitivity occurring throughout the day. Because of this, disrupted sleep may reduce the effectiveness of pain management and delay recovery. In future research, electronic sleep and pain diaries would make it possible to collect additional data regarding the variability in pain sensitivity throughout the day, which, in turn, may contribute to optimizing pharmacological treatments. Additionally, because of the demonstrated link between sleep and pain, it seems appropriate to strive for a complementary therapy approach that includes sleep problems as an important treatment target in multimodal pain management programs for people in pain [66]. Pilot studies combining pain and sleep management in the form of cognitive behavioral therapy have already shown promising results of a hybrid therapy [67,68]. Future studies will have to show whether this approach is effective in any pain condition or only in subgroups with high pain intensity and interference.

Subjectively measured sleep quality and insomnia severity did not seem to be associated with pain duration; however, one study found sleepiness to be negatively associated with the duration of pain [55]. Important in the interpretation of this result is that only patients with pain that had been present for at least three months were included. This implies that patients in the early phase of chronic pain seem to report more sleep problems than patients who are suffering from pain over a longer period. It appears more logical not to find any association. After all, in an earlier phase, spinal pain can be the cause of the sleep problems, both by the physical aspect of pain and by mental arousal. Because of sleep disturbances, certain negative cognitions and behaviors around sleep and pain can subsequently arise, and these are perpetuating factors that sustain the vicious cycle between sleep and pain in the long term [69].

### Strengths and Limitations

Several limitations and strengths of the included studies and the review itself need to be considered when interpreting the results from this review.

Throughout the various studies included, besides the lack of an unambiguous definition of insomnia, many different questionnaires and accompanying cut-off scores were used to investigate sleep. Furthermore, only one study used actigraphy to objectively assess sleep based on detected activity counts [42]. However, data obtained via polysomnography, which is considered the gold standard for assessing sleep [70], were not used in any of the eligible studies. Future research should include both objective and subjective assessment tools to optimize the interpretation of the variation in sleep data [71]. In addition to sleep, no clear delineation of chronic spinal pain was made, so a diversity of spinal pathologies, both specific and non-specific, was included in this review. Possible differences in results could therefore be caused by the specificity of a subgroup of spinal pain regarding the underlying physiological processes.

This review focused on pain intensity, but in order to maintain an overview between the various assessment tools used in the included studies, pain severity was also included under this term, although it covers a wider range of facets such as pain interference, catastrophizing, and beliefs [72]. Indeed, chronic pain is a complex experience that encompasses sensory, emotional, cognitive, and social components [45,58]. The assumption of purely capturing pain intensity in the presented results can therefore not be substantiated. Expanding the research question and exploring the multifactorial processes associated with pain would provide a broader interpretation of the association between sleep and CSP parameters.

Checking for age and sex as well as for psychological factors such as anxiety and depression as confounders was incorporated into the screening for risk of bias. However, parameters such as medication use and comorbidities such as obesity may also have impacted the results and thus should have been taken into account. Regarding this first parameter, in fact, it is not only sleep medication that affect sleep quality. Several other types of drugs such as analgesics (e.g., opioids), cardiovascular medications, antidepressants, corticosteroids, and statins can interfere with sleep [63,73,74]. In clinical practice, pharmacological approaches should therefore be adjusted in terms of dosing and timing to minimize sleep disturbances [63]. In addition, the timing of possible pain medication intake may have influenced the findings of more severe pain in the morning compared to afternoon or evening [40,45]. Thus, due to a lack of information on medication intake, these results are open to interpretation as subjects may have taken analgesics after the morning assessment. Furthermore, overweight and obesity, frequently reported by body mass index (BMI), are assumed to be risk factors for low back pain [75]. Mechanisms underlying this association can be attributed to increased mechanical load on the spine, structural degenerations of the disc and vertebral endplate, and chronic inflammation [75].

By only including articles published after January 2009, potentially previously published relevant articles on CNP were not included in the review. Since only a limited number of studies on neck pain were found, extending the search strategy to articles published before January 2009 might have led to stronger findings. By not including gray literature, some publication bias may also have occurred. Additionally, the research question in this systematic review differs somewhat from the one of Kelly et al. (2011) by focusing more specifically on the association between sleep and spinal pain, including longitudinal cohort studies on baseline healthy individuals [16]. However, due to the fact that most of the recorded studies were cross-sectional, it remains difficult to draw firm conclusions on causality.

Due to the considerable differences in patient selection, the definition of exposure and outcome parameters, the study protocols, and the assessment tools without consistent cut-off scores and time periods, the ability to synthesize outcome measures in a meta-analysis seems limited and of less relevance. Uniformity of sleep assessment is an important step to increase the ease and accuracy of treatment comparisons across studies.

In addition to the limitations, some strengths should be mentioned. This review added patients with chronic neck pain to the study population in examining the association between sleep and pain, which had not been examined to date. Furthermore, a systematic approach based on the PRISMA guidelines was applied. In adopting this approach, the different phases of conducting the review (i.e., literature screening, risk of bias scoring, collection of results) were performed by two independent researchers, thus limiting detection and performance bias.

## 5. Conclusions

The current systematic review indicates that weak to moderate evidence is present for an association between sleep and spinal pain. In this relationship, sleep seems to be a stronger predictor for the development of CSP than vice versa. Addressing the frequently reported sleep problems in chronic back pain patients is therefore a necessary complement to pain management in order to provide an optimal treatment outcome in this population.

## Figures and Tables

**Figure 1 jcm-10-03836-f001:**
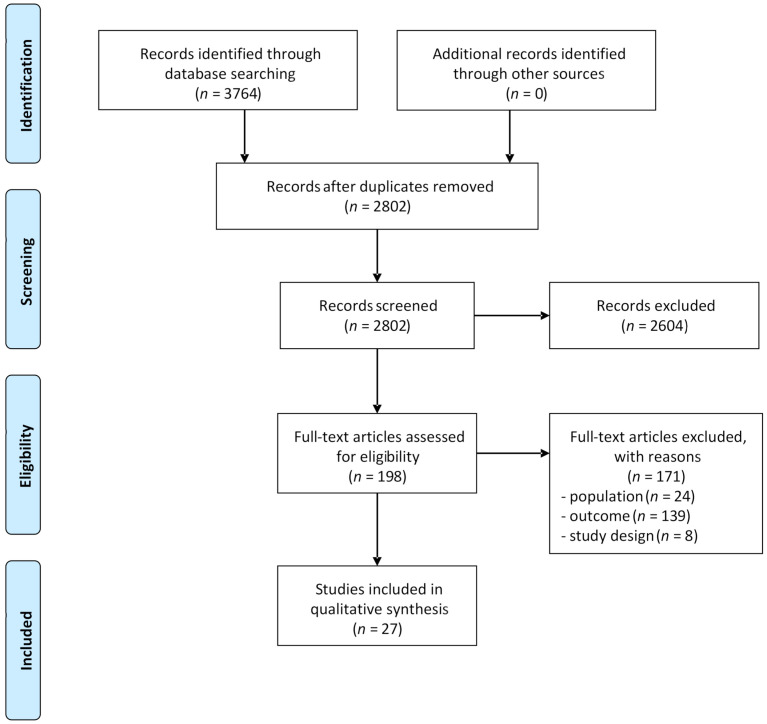
Flowchart.

**Table 1 jcm-10-03836-t001:** Free text words used to build the search strategy.

Population		AND	Outcome
chronic	back pain/ache		sleep
persistent	neck pain/ache		insomnia
lasting	cervical pain/ache		agrypnia
longterm	spinal pain/ache		
long term	vertebral pain/ache		
	lumbar pain/ache		
	lumbosacral pain/ache		
	backpain		
	backache		
	dorsalgia		
	lumbago		
	lumbalgia		
	lumbodynia		
	whiplash		
	fbss		
	failed back surgery		
	failed back syndrome		
	postlaminectomy syndrome		
	post-laminectomy syndrome		
	cervicobrachial neuralgia/pain/syndrome/disease/radiculopathy/compression syndrome		
	cervico-brachial neuralgia/pain/syndrome/disease/radiculopathy/compression syndrome		
	cervical neuralgia/syndrome/disease/radiculopathy/compression syndrome		
	neck shoulder arm syndrome		
	shoulder arm neck syndrome		

**Table 2 jcm-10-03836-t002:** Methodological quality of cohort studies.

Study	Selection				Comparability		Exposure			Level of Evidence
	1	2	3	4	5	6	7	8	9	
Agmon et al. [33]	−	+	−	−	+	−	−	+	−	B
Kääria et al. [34]	−	+	−	−	+	+	−	+	+	B
Mork et al. [35]	+	+	−	−	+	+	−	+	−	B
Pakpour et al. [36]	−	+	−	−	+	+	−	+	+	B
Skarpsno et al. [37]	+	+	−	−	+	+	−	+	−	B
Uhlig et al. [38]	+	+	−	−	+	+	−	+	−	B

Newcastle–Ottawa quality assessment scale: cohort studies. Legend: +, score fulfilled; −, score not fulfilled/too little information; 1, representativeness of exposed cohort; 2, selection of non-exposed cohort; 3, ascertainment of exposure; 4, demonstration that outcome of interest was not present at start of study; 5, study controls for age and sex; 6, study controls for psychological factor(s); 7, assessment of outcome; 8, follow-up > 3 months; 9, <20% loss to follow-up.

**Table 3 jcm-10-03836-t003:** Methodological quality of case-control Studies.

Study	Selection				Comparability		Exposure			Level of Evidence
	1	2	3	4	5	6	7	8	9	
Häggmann-Henrikson et al. [39]	−	+	−	+	+	−	−	+	−	B
Heffner et al. [40]	−	+	+	+	+	+	−	+	−	B
Hong et al. [41]	+	−	+	+	+	−	−	+	−	B
O’Donoghue et al. [42]	−	−	−	+	+	−	+	+	−	B
Sezgin et al. [43]	−	−	−	+	+	−	−	+	−	B

Newcastle-Ottawa quality assessment scale: case-control studies. Legend: +, score fulfilled; −, score not fulfilled/too little information; 1, adequate case definition; 2, representative cases; 3, selection of controls; 4, definition of controls; 5, study controls for age and sex; 6, study controls for psychological factor(s); 7, ascertainment of exposure; 8, same method of ascertainment for cases and controls; 9, non-response rate.

**Table 4 jcm-10-03836-t004:** Methodological quality of cross-sectional studies.

Study	Selection				Comparability		Outcome		Level of Evidence
	1	2	3	4	5	6	7	8	
Artner et al. [11]	−	−	−	+	+	−	−	+	C
Bahouq et al. [12]	−	−	−	+	+	−	−	+	C
Burgess et al. [44]	+	+	−	+	−	+	−	+	C
Gerhart et al. [45]	+	−	−	+	+	−	−	+	C
Ho et al. [46]	+	+	−	+	+	+	−	+	C
Juan et al. [47]	−	−	−	+	−	−	−	+	C
Kennedy et al. [48]	−	−	−	+	+	−	−	+	C
Kim et al. [49]	−	−	−	+	+	+	−	+	C
Kim et al. [50]	−	−	−	+	+	−	−	+	C
Ouchi et al. [51]	−	−	+	+	+	+	−	+	C
Purushothaman et al. [13]	−	−	−	−	−	−	−	−	C
Rodrigues-De-Souza et al. [52]	−	+	−	+	+	+	−	+	C
Shmagel et al. [53]	+	−	−	+	+	−	−	+	C
Srivastava et al. [54]	−	−	−	−	+	+	−	+	C
Uchmanowicz et al. [55]	−	−	−	−	+	−	−	+	C
Wang et al. [56]	−	−	−	−	+	+	−	+	C
Yamada et al. [57]	−	+	−	+	+	+	−	+	C

Newcastle-Ottawa quality assessment scale: adapted for cross-sectional studies. Legend: +, score fulfilled; −, score not fulfilled/too little information; 1, representative sample; 2, sample size; 3, non-respondents; 4, ascertainment of exposure; 5, study controls for age and sex; 6, study controls for psychological factor(s); 7, assessment of outcome; 8, statistical test.

**Table 5 jcm-10-03836-t005:** Level of evidence according to the 2005 classification system of the Dutch Institute for Healthcare Improvement CBO.

	Intervention
A1	Systematic review of at least two conducted independent of each other were of evidence level A2
A2	Randomized double-blinded comparative clinical research of good quality and efficient size
B	Comparative research but not with all characteristics as mentioned for A2. This includes also patient-control research and cohort research
C	Not comparative research
D	Opinion of experts

**Table 6 jcm-10-03836-t006:** Strength of conclusion.

	Conclusion Based On
1	Research of evidence level A or at least two independently conducted studies of evidence level A
2	One research work of evidence level A2 or at least 2 independently conducted studies of evidence level B
3	One research work of evidence level B or C
4	Opinion of experts or inconclusive or inconsistent results between various studies

**Table 7 jcm-10-03836-t007:** Evidence Table.

Study	Subjects	Outcome Measures		Association	Main Results
Author Design	Sample Characteristics	Pain	Sleep	Between Sleep and Pain	Correlation Coefficient (r), Regression Coefficient (β or B),Odds Ratio (OR), Risk Ratio (RR)
Agmon et al. [33] Cohort study	Healthy subjects (A) No CBP at T1 and T2 *n* = 1356 Age 45.45 [8.50] ♀ 24% (B) No CBP at T1 *n* = 1527 Age 45.55 [8.53] ♀ 26%	Medical interview (CBP ≥ 3 m) T1: baseline T1 → T2: 18 m (range 11–19) T2 → T3: 17 m (range 11–29)	AIS (insomnia)	(A) 1. Insomnia at T1 → CBP at T3 relative to T2 2. Insomnia at T2 → CBP at T3 relative to T2 3. Change in insomnia from T1 to T2 → change in CBP from T2 to T3 (B) 1. Change in CBP from T1 to T2 → change in insomnia from T2 to T3	(A) 1. No association: No insomnia: OR = 1 Insomnia: OR = 0.942, 95% CI = 0.72–1.24, NS (Model 1) 2. ↑ odds of CBP at T3 in patients with insomnia at T2: No insomnia: OR = 1 Insomnia: OR = 1.36, 95% CI = 1.27–1.51, *p* < 0.05 (Model 1) 3. ↑ odds of CBP at T3 relative to T2 in patients with ↑ insomnia severity from T1 to T2: No insomnia: OR =1 Insomnia: OR = 1.40, 95% CI = 1.10–1.71 (Model 2) (B) 1. No association: β = 0.02, NS Model 1: unadjusted Model 2: adjusted for age, gender, education level, PA intensity, self-rated health, smoking status, BMI, CRP, and time of FU
Artner et al. [11] Cross-sectional study	CLBP/CNP *n* = 1016 Age 57.07 [14.28] ♀ 56%	NRS (pain intensity)	ODQ (sleep deprivation intensity)	1. Sleep deprivation intensity ~ pain intensity subgroups 2. Sleep deprivation intensity ~ pain intensity	1. Difference between subgroups NRS < 5 vs. NRS ≥ 5: χ^2^ = 54.716, r = 0.303, *p* < 0.001 NRS < 5 vs. NRS > 7: χ^2^ = 65.664, r = 0.474, *p* < 0.001 2. + association: β = 0.236, *p* < 0.00001
Bahouq et al. [12] Cross-sectional study	CLBP *n* = 100 Age 43.28 [7.5] ♀ 50%	VAS (pain intensity)	ISI (insomnia)	1. Insomnia ~ pain intensity 2. Insomnia ~ CLBP duration 3. Pain intensity → insomnia	1. + correlation: r = 0.587, *p* < 0.0001 2. No correlation: r = 0.001, NS 3. + association: β = 1.984, 95% CI = 1.517–2.451, *p* < 0.0001
Burgess et al. [44] Cross-sectional study	CLBP *n* = 87 Age 40.0 [11.49] ♀ 64.4%	MPQ-SF (pain ratings)	PROMIS—SF8a (sleep disturbance)	1. Sleep disturbance ~ pain intensity	1. + correlation Sensory pain ratings: r = 0.44, *p* < 0.05 Affective pain ratings: r = 0.39, *p* < 0.05 VAS pain intensity: r = 0.51, *p* < 0.05
Gerhart et al. [45] Cross-sectional study	CLBP *n* = 105 Age 46.30 [12.1] ♀ 48.6%	Self-reported questionnaire (pain intensity) 14 days, 5x/day between 8.50 a.m. and 8.50 p.m.	Self-reported questionnaire (sleep quality)	1. Sleep quality ~ pain intensity 2. Prior night sleep quality → subsequent pain intensity 3. Prior night sleep quality → hour by hour pain intensity 4. Prior day pain intensity → subsequent sleep quality	1. Correlation: r = −0.35, *p* < 0.01 2. Association: B = −0.16, *p* ≤ 0.00001 3. Association: 8:50 a.m.: B = −0.42, *p* < 0.00065 11:50 a.m.: B = −0.15, *p* < 0.01 No association: 2:50 p.m.: B = −0.07, NS 5:50 p.m.: B = −0.04, NS 8:50 p.m.: B = −0.09, NS 4. No association: NS Model adjusted for gender, age, employment, and disability compensation
Häggman-Henrikson et al. [39] Case-control study	Cases: CWAD *n* = 50 Age 39 [11] ♀ 78% Controls: Healthy subjects *n* = 50 Age- and sex-matched		Self-reported questionnaire (sleep problems)	1. CWAD → sleep problems	1. ↑ odds of sleep problems in CWAD: Controls: OR = 1 CWAD: OR = 15.58, 95% CI = 5.51–44.06, *p* < 0.0001 *
Heffner et al. [40] Case-control study	Cases: CLBP *n* = 25 Age 30.82 [11.38] ♀ 60% Controls: Healthy subjects *n* = 25 Age- and sex-matched	MPQ-SF(pain ratings)	PSQI(sleep quality)	1. Sleep quality past month ~ morning pain ratings in CLBP 2. Sleep quality past night ~ morning pain ratings in CLBP	1. + correlation in CLBP: - Sensory pain ratings: r = 0.43, *p* = 0.03 - Affective pain ratings: r = 0.56, *p* = 0.004 - MPQ-SF total score: r = 0.51, *p* = 0.01 2. + correlation in CLBP: - Affective pain ratings: r = 0.47, *p* = 0.02 No correlation in CLBP:- Sensory pain ratings: NS- MPQ-SF total score: NS
Ho et al. [46] Cross-sectional study	Health survey *n* = 30,699Age 52.2 [15.2]♀ 54.3%	Self-reported questionnaire(CLBP)	Self-reported questionnaires(insomnia)	1. Insomnia → CLBP	1. ↑ odds of CLBP in patients with insomnia: No insomnia: OR = 1Insomnia: OR = 1.99, 95% CI = 1.79–2.21, *p* < 0.001 Model adjusted for age, BMI, sex, smoking, PA, depression, anxiety, and osteoarthritis
Hong et al. [41] Case-control study	Cases: CLBP *n* = 47Age 63.38 [9.55]♀ 59.6%Controls: Healthy subjects *n* = 44Age 63.64 [4.7] ♀ 63.6%	SF-36 bodily pain scale(bodily pain)	PSQI(sleep quality)	1. Sleep quality ~ pain duration 2. CLBP → sleep disturbance	1. No correlation: r = 0.015, NS 2. No association: Controls: OR = 1 CLBP: OR = 1.01, 95% CI = 0.43–2.37, NS *
Juan et al. [47] Cross-sectional study	CNP *n* = 231 Age 48.9 [13.9] ♀ 71.0%	VAS (pain intensity)	PSQI (sleep quality)	1. Sleep quality ~ pain intensity 2. Pain intensity → sleep quality	1. - correlation: r = 0.15, *p* < 0.05 2. EE = 0.136, CR = 1.994, *p* = 0.046
Kääria et al. [34] Cohort study	Health survey No CNP at T1 *n* = 5277 ♀ 80.7%	Self-reported questionnaire (CNP) T1: baseline T2: 5–7 y	4-item Jenkins Sleep Questionnaire (sleep problems)	1. Rare/occasional sleep problems at T1 → CNP at T2 2. Frequent sleep problems at T1 → CNP at T2	1. ♀ ↑ odds of CNP in women with rare/occasional sleep problems: No sleep problems: OR = 1 OR = 1.28, 95% CI = 1.05–1.55 (Model 1); NS in Model 2 and 3 ♂ No association: NS in Model 1, 2, and 3 (No sleep problems: OR = 1) 2. ♀ ↑ odds of CNP in women with frequent sleep problems: No sleep problems: OR = 1 OR = 1.54, 95% CI = 1.22–1.95 (Model 3) ♂ ↑ odds of CNP in men with frequent sleep problems: No sleep problems: OR = 1 OR = 1.91, 95% CI = 1.10–3.33 (Model 1) NS in Model 2 and 3 ♀ /♂ Model 1: adjusted for age ♀ Model 2: adjusted for age, physical workload, emotional exhaustion, bullying, GHQ, sleep problems, acute NP, LBP, and BMI ♂ Model 2: adjusted for age, occupational class, emotional exhaustion, sleep problems, acute NP, and LBP ♀ Model 3: adjusted for age, bullying, sleep problems, acute NP, LBP, and BMI ♂ Model 3: adjusted for age, occupational class, emotional exhaustion, acute NP, and LBP
Kim et al. [49] Cross-sectional study	CLBP *n* = 481 Age 58.2 [16.7] ♀ 59%	NRS (pain intensity) Self-reported questionnaire (pain duration)	ISI (insomnia)	1. Insomnia ~ pain duration 2. Pain intensity → insomnia	1. No association: NS 2. ↑ odds of insomnia in patients with high pain intensity: <7 NRS: OR = 1 ≥7 NRS: OR = 2.342, 95% CI = 1.257–4.365, *p* = 0.007 Model adjusted for age, gender, pain intensity, comorbid neuropathic and musculoskeletal pain, anxiety, and depression
Kim et al. [50] Cross-sectional study	CNP *n* = 218 Age 52.8 [14.3] ♀ 56.9%	NRS (pain intensity) Self-reported questionnaire (pain duration)	ISI (insomnia)	1. Insomnia ~ pain duration 2. Insomnia ~ pain intensity 3. Pain duration → clinical insomnia 4. Pain intensity → clinical insomnia	1. No correlation: R = 0.094, NS 2. + correlation: R = 0.350, *p* < 0.001 3. No association: <1 y: OR = 1 ≥1 y: OR= 1.469, 95% CI 0.778–2.772), NS 4. ↑ odds of insomnia in patients with high pain intensity: <7 NRS: OR = 1 ≥7 NRS: OR = 2.457, 95% CI = 1.119–5.395, *p* = 0.025 Model adjusted for pain intensity, comorbid neuropathic and musculoskeletal pain, anxiety, and depression
Mork et al. [35] Cohort study	Health survey No CLBP/CNP at T1 *n* = 26,896 ♀ 50.2%	Self-reported questionnaire (CLBP and CNP/shoulder pain) T1: baseline T2: 11 y (range 9–13)	Self-reported questionnaire (sleep problems)	1. Sleep problems at T1 → CLBP at T2 2. Sleep problems at T1 → CNP at T2	1. ↑ risk of CLBP in patients with sometimes or often/always sleep problems: Never: RR = 1 ♀ Sometimes: RR = 1.32, 95% CI = 1.22–1.43 ♂ Sometimes: RR = 1.30, 95% CI = 1.18–1.43 ♀ Often/always: RR = 1.66, 95% CI = 1.41–1.95 ♂ Often/always: RR = 1.51, 95% CI = 1.20–1.91 2. ↑ risk of CNP in patients with sometimes or often/always sleep problems: Never: RR = 1 ♀ Sometimes: RR = 1.31, 95% CI = 1.24–1.40 ♂ Sometimes: RR = 1.23, 95% CI = 1.15–1.33 ♀ Often/always: RR = 1.53, 95% CI = 1.35–1.74 ♂ Often/always: RR = 1.58, 95% CI = 1.35–1.86 Model adjusted for age, BMI, physical exercise, psychological well-being, smoking status, and occupation
O’Donoghue et al. [42] Case-control study	Cases: CLBP *n* = 15 Age 45.0 [11.53] ♀ 53% Controls: Healthy subjects *n* = 15 Age 38.46 [10.57] ♀ 53%	SF36-v2 bodily pain scale (pain intensity)	PSQI (sleep quality) ISI (insomnia) Actiwatch (TST, WASO, SOL, SE)	1. Objective sleep~pain intensity 2. Sleep quality~pain intensity 3. Insomnia~pain intensity	1. Correlation in CLBP: no description of correlation coefficient or *p*-value 2. Correlation in CLBP: r = -.59, *p* = 0.021 (strong correlation) 3. Correlation in CLBP: r = -.74, *p* = 0.001 (strong correlation)
Ouchi et al. [51] Cross-sectional study	Health survey CLBP *n* = 286 Age 40.5 [10.8] ♀ 46,5%	JOABPEQ (CLBP)	Self-reported questionnaire (sleep duration)	1. CLBP → sleep duration	1. No association with sleep duration of 7 to 8 h and ≥8 h ↑ odds of a sleep duration of <6 h in CLBP patients: 6–7 h: OR = 1 <6 h: OR = 1.402, 95% CI = 1.009–1.947, *p* = 0.044 7–8 h: OR 1.460, 95% CI = 0.974–2.188, NS ≥8 h: OR = 0.614, 95% CI = 0.320–1.178, NS
Pakpour et al. [36] Cohort study	CLBP *n* = 761 Age 41.15 [12.24] ♀ 44.6%	VAS (pain intensity, recovery) T1: baseline T2: 6 m	PSQI (sleep quality)	1. Sleep problems at T1 → pain intensity at T2 2. Developing sleep problems → recovery from CLBP 3. Persistent sleep problems → recovery from CLBP 4. Resolving sleep problems → recovery from CLBP	1. ↑ odds of pain intensity at T2 in patients with sleep problems at T1/ No sleep problems at T1: OR = 1 Sleep problems: OR = 2.48, 95% CI = 1.62–3.70 2. ↑ odds of non-recovery in terms of pain intensity for developing sleep problems: No sleep problems at T1 and T2: OR = 1 Developing sleep problems: OR = 2.88, 95% CI = 1.32–6.31 3. ↑ odds of non-recovery in terms of pain intensity for persistent sleep problems: No sleep problems at T1 and T2: OR = 1 Persistent sleep problems: OR = 3.45, 95% CI = 1.59–7.46 4. ↓ odds of non-recovery in terms of pain intensity for resolving sleep problems No sleep problems at T1 and T2: OR = 1 Resolving sleep problems: OR = 0.49, 95% CI = 0.26–0. 93 Model adjusted for baseline depressive symptoms, baseline pain intensity and duration, anxiety, age gender, BMI, and occupational status
Purushotaman et al. [13] Cross-sectional study	CBP *n* = 120 Age 55 ♀ 52.5%	NRS (pain intensity)	ISI (insomnia)	1. Insomnia ~ pain intensity	1. + correlation: r = 0.38, *p* < 0.001
Rodrigues-De-Souza et al. [52] Cross-sectional study	CLBP Spain *n* = 30 Age 51.1 [13] ♀ 87.5 Brazil *n* = 30 Age 49.7 [10.5] ♀ 76.5	NPRS (pain intensity) PRI (MPQ) (pain ratings) PPI (MPQ) (pain intensity)	PSQI (sleep quality)	1. Sleep quality ~ pain intensity in Brazilian subjects 2. Sleep quality ~ pain intensity in Spanish subjects	1. No correlation for NPRS, PRI, PPI: NS 2. + correlation NPRS: r = 0.364, *p* < 0.05 (weak correlation) PRI: r = 0.486, *p* < 0.01 (moderate correlation) No correlation for PPI: NS
Sezgin et al. [43] Case-control Study	Cases: CLBP *n* = 200 Age 50.2 [14.2] ♀ 50% Controls: Healthy subjects *n* = 200 Age 49.7 [13.6] ♀ 50%	MPQ-SF (pain intensity) Self-reported questionnaire (pain duration)	PSQI (sleep quality)	1. CLBP → sleep quality 2. Sleep quality ~ pain duration 3. Sleep quality ~ pain intensity	1. ↑ odds of poor sleep quality (PSQI > 5) in CLBP: Controls: OR = 1 CLBP: OR = 4.54, 95% CI = 2.98–6.91, *p* < 0.0001 * 2. ↑ daytime dysfunction in patients with pain duration > 11 y: *p* = 0.02 3. + correlation: Sensory pain ratings: r = 0.47, *p* < 0.001 Affective pain ratings: r = 0.35, *p* < 0.001 MPQ-SF total score: r = 0.47, *p* < 0.001 VAS: r = 0.34, *p* < 001 ETPI: r = 0.35, *p* < 001
Shmagel et al. [53] Cross-sectional study	Health Survey *n* = 5103 ♀ 50.8%	Self-reported questionnaire (CLBP)	Self-reported questionnaire (sleep disturbance)	1. CLBP → sleep disturbances	1. ↑ odds of sleep disturbances in CLBP patients: No CLBP: OR = 1 CLBP: OR = 3.90; 95% CI = 3.22–4.73, *p* < 0.0001 Model adjusted for age, race, gender, and education
Skarpsno et al. [37] Cohort study	CLBP *n* = 6200 ♀ 59.9%	Self-reported questionnaire (CLBP) T1: baseline T2: ± 10 y	Self-reported questionnaire (sleeplessness) Self-reported questions (insomnia)	1. Sleeplessness at T1 → recovery from CLBP at T2 2. Number of insomnia symptoms at T1 → recovery from CLBP at T2	1. ↑ risk of non-recovery from CLBP in ♀ and ♂ when experiencing sleeplessness often/always: Never/seldom: RR = 1 ♀ Often/always: RR = 0.65, 95% CI = 0.57–0.74 ♂ Often/always: RR = 0.81, 95% CI = 0.69–0.95 2. ↑ risk of non-recovery from CLBP in ♀ with ↑ number of insomnia symptoms No symptoms: RR = 1 ♀ 1 symptom: RR = 0.81, 95% CI = 0.72–0.91 ♀ 2 symptoms: RR = 0.68, 95% CI = 0.57–0.80 ♀ 3 symptoms: RR = 0.60, 95% CI = 0.46–0.77 No association in ♂: ♂ 1 symptom: RR = 0.99, 95% CI = 0.89–1.10 ♂ 2 symptoms: RR = 0.84, 95% CI = 0.71–1.00 ♂ 3 symptoms: RR = 0.82, 95% CI = 0.59–1.14
Srivastava et al. [54] Cross-sectional	CLBP *n* = 100 ♀ 64%	Self-reported (pain duration)	PSQI (sleep quality)	1. Sleep quality ~ pain duration	1. No difference in sleep quality between pain duration ≤ 1 year and > 1 year: *p* = 0.06
Uchmanowicz et al. [55] Cross-sectional study	CLBP *n* = 100 Age 49.53 [10.92] ♀ 62%	VAS (pain intensity)	AIS (insomnia) ESS (sleepiness)	1. Insomnia ~ pain intensity 2. Insomnia ~ pain duration 3. Sleepiness ~ pain intensity 4. Sleepiness ~ pain duration 5. pain intensity → insomnia 6. pain duration → insomnia 7. pain duration → sleepiness	1. + correlation: r = 0.462, *p* < 0.001 (weak correlation) 2. − correlation: r = −0.335, *p* = 0.001 (weak correlation) 3. + correlation: r = 0.233, *p* = 0.02 (very weak correlation) 4. - correlation: r = −0.307, *p* = 0.002 (weak correlation) 5. Pain intensity is an independent predictor of insomnia: R = 1.515, *p* < 0.001 6. Pain duration is an independent predictor of insomnia: R = −0.303, *p* = 0.009 7. Pain duration is an independent predictor of sleepiness: R = −0.247, *p* = 0.014
Uhlig et al. [38] Cohort study	Health Survey No CLBP/CNP at T1 No insomnia at T1 *n* = 13,113 Age 43.4 [12.2] ♀ 54.8% Insomnia at T1 *n* = 316 Age 44.5 [12.2] ♀ 57.3%	Self-reported questionnaire (CLBP and CNP) T1: baseline T2: 11 y (range 9–13)	Self-reported questionnaire (insomnia)	1. Insomnia at T1 → CLBP at T2 2. Insomnia at T1 → CNP at T2	1. ↑ risk of CLBP in T1 insomnia: No insomnia: RR = 1 Insomnia: RR = 1.36, 95% CI = 1.11–1.68 2. ↑ risk of CNP in T1 insomnia: No insomnia: RR = 1 Insomnia: RR = 1.34, 95% CI = 1.10–1.63 Model adjusted for sex, age, BMI, physical activity, education, HADS, and smoking
Wang et al. [56] Cross-sectional study	CLBP *n* = 225 Age 40.7 [11.4] ♀ 45.8%	VAS (pain intensity)	ISI (insomnia)	1. Insomnia~severe CLBP 2. Severe CLBP → insomnia	1. + correlation: r = 0.33, *p* < 0.01 2. ↑ odds of clinical insomnia in severe CLBP compared to no severe CLBP: No severe CLBP: OR = 1 Severe CLBP: OR = 2.80, 95% CI = 1.52–5.17, *p* = 0.001
Yamada et al. [57] Cross-sectional study	Health survey *n* = 22,948 Age 52.1 [9.8] ♀ 96.8%	Self-reported questionnaire (CLBP)	Self-reported questionnaire (sleep sufficiency and duration)	1. sleep sufficiency → CLBP 2. sleep duration → CLBP	1. ↑ odds of CLBP in persons with lower sleep sufficiency: More than enough sleep: OR = 1 (Model 1) Enough sleep: OR = 1.68, 95% CI = 1.52–1.86, *p* < 0.001 Not enough sleep: OR = 2.65, 95% CI = 2.36–2.97, *p* < 0.001 No sleep at all: OR = 4.58, 95% CI = 3.62–5.81, *p* < 0.001 2. ↑ odds of CLBP in persons with sleep duration of <5 h and ≥5 h to <6 h: ≥6 h to <7 h: OR = 1 (Model 2) <5 h: OR = 1.44, 95% CI = 1.30–1.60, *p* < 0.001 ≥5 h to <6 h: OR = 1.11, 95% CI = 1.03–1.19, *p* < 0.01 ↓ odds of CLBP in persons with sleep duration of ≥7 h to <8 h: ≥6 h to <7 h: OR = 1 (Model 2) ≥7 h to <8 h: OR = 0.89, 95% CI = 0.80–0.98, *p* < 0.05 No association for sleep duration of ≥7 h to <8 h and ≥9 h: ≥8 h to <9 h: OR = 0.91, 95% CI = 0.75–1.12, NS ≥9 h: OR = 0.77, 95% CI = 0.44–1.33, NS Model 1: adjusted for age, sex, BMI, regular exercise, smoking, employment status, mood, anxiety, and sleep sufficiency Model 2: adjusted for age, sex, BMI, regular exercise, smoking, employment status, mood, and anxiety

Subjects: C(L)BP, chronic (low) back pain; CNP, chronic neck pain; CWAD, chronic whiplash associated disorders, *n*, number of subjects; ♀, percentage female subjects; ♂, percentage male subjects; Time: a.m., ante meridiem; FU, follow-up; h, hour(s); m, month(s); p.m., post meridiem; T, time point; y, year(s); Pain: ETPI, evaluative total pain intensity; MPQ(-SF), McGill Pain Questionnaire (-Short Form); N(P)RS, Numeric (Pain) Rating Scale, PPI, Present Pain Index; PPR, Pain Rating Index; JOABPEQ, Japanese Orthopedic Association Back Pain Evaluation Questionnaire; VAS, Visual Analogue Scale; Sleep: AIS, Athens Insomnia Scale; ESS, Epworth Sleepiness Scale; ISI, Insomnia Severity Index; ODQ, Oswestry Disability Questionnaire; PSQI, Pittsburgh Sleep Quality Index; PROMIS-SF8a, Patient-Reported Outcomes Measurement Information System—Short Form; SE, sleep efficiency; PSQI, Pittsburgh Sleep Quality Index; SOL, sleep onset latency; TST, total sleep time; WASO, wake after sleep onset; Results: CI, confidence interval; CR, critical ratio; EE, estimated effect; NS, not significant; OR, odds ratio; r, correlation coefficient; RR, risk ratio, ↑, increased/higher; ↓, decreased/lower; *, odds ratios are self-calculated based on given data; Adjusted models: BMI, body mass index; CRP, C-reactive protein; PA, physical activity.

**Table 8 jcm-10-03836-t008:** Overview of the details of the used sleep questionnaires.

	Outcome	Assessment Tool	Construct	Time Span	Interpretation	Cut-Off Score	
SLEEP	Sleep quality	Pittsburgh Sleep Quality Index (PSQI) [36,40,41,42,43,46,52,56]	24 items 7 dimensions Total score ranging from 0 to 21	Past month Past night [40]	↑ scores = ↓ sleep quality	≥5 [36] >5 [41,42] >6 [40] >8 [52]	Clinical sleep disturbance Sleep disturbance Clinical sleep disturbance Poor sleep quality
		Self-reported questionnaire [52] *Rate the overall quality of your sleep.*	5-point Likert scale	past night	0—not at all restful 1—a little restful 2—somewhat restful 3—very restful 4—extremely restful		
	Insomnia	Insomnia Severity Index (ISI) [12,13,42,48,49,50,54,56]	7 items 5-point Likert scale Total score ranging from 0 to 28	Past 2 weeks	↑ scores = ↑ insomnia severity 0–7 no clinically significant insomnia 8–14 sub-threshold insomnia [12,13,42,49,50,56] 15–21 clinical insomnia (moderate severity) [12,13,42,49,50,56] 22–28 clinical insomnia (severe)	14 [13,59] >15 [57] ≥15 [49,56]	Primary insomnia Clinical insomnia Clinical insomnia
		Athens Insomnia Scale (AIS) AIS-5 [33] AIS-8 [47]		Past month	↑ scores = ↑ insomnia severity	≥6 [47]	Insomnia
		Self-reported question(s) *How often during the last 3 months have you:* *(1) Had difficulty falling asleep at night?* *(2) Woken up repeatedly during the night?* *(3) Woken too early and could not get back to sleep?* *(4) Felt sleepy during the day?* [55] *(1) Have you had problems falling asleep during the last month,* *(2) During the last month, did you ever wake up too early, not being able to fall asleep again?* *(3) During the last year, have you been troubled by sleepiness to such a degree that it affected your work?* [37,38]		Past 3 months Past month	(1–3) never/seldom, sometimes, several times a week (1–2) never, occasionally/sometimes, often, almost every night (3) yes, no		Insomnia if person answered several times a week to question 4 and at least one of questions 1–3; Insomnia if person answered often or almost every night to questions 1 and 2 or yes to question 3
	Sleep disturbance/problems/sleeplessness	PROMIS sleep disturbance—short form 8a [51]	5-point Likert scale	Past week	↑ scores = ↑ sleep disturbance	T score > 50	disturbed sleep
		4-item Jenkins Sleep Questionnaire (JSQ) [34]	6-point Likert scale	Past 4 weeks	↑ scores = ↑ sleep disturbance 1—not at all 2—1 to 3 days 3—4 to 7 days 4—8 to 14 days 5—15 to 21 days 6—22 to 28 days		Rare to occasional sleep problems if any of the problems occurred 1–14 times; frequent sleep problems if any of the problems occurred ≥ 15 times
		Self-reported question *During the last month, have you had any problems falling asleep or sleep problems?* [35] *Sleep problems* [39] *Have you ever told a doctor or other health professional that you have trouble sleeping?* [45] *How often do you suffer from sleeplessness?* [37]		Past month	(1) never, sometimes, often, almost every night 0—no, never 1—yes, seldom, every year 2—yes, often, every month 3—yes, very often, every week 4—yes, always, every day 1—never, or just a few times a year 2—1–2 times a month 3—approximately once a week 4—more than once a week	1 or 2 3 or 4	Low frequency Frequent
	Sleep deprivation intensity	Oswestry Disability Questionnaire (ODQ) [11]	10 items 6-point Likert scale		No disturbance No disturbance when taking analgesics Sleep < 6 h even when using analgesics Severe sleep deprivation (<4 h) due to pain, even when using analgesics		
	Sleep sufficiency	Self-reported questionnaire *Sleep sufficiency* [60]		Past month	Not at all Not enough Enough More than enough		
	Sleep duration	Self-reported questionnaire *Sleep duration* [44] *Average sleep duration* [50]		Past month	<6 h per night 6–7 h per night 7–8 h per night ≥8 h per night <5 h ≥5 to <6 h ≥6 to <7 h ≥7 to <8 h ≥8 to <9 h ≥9 h		
	Daytime sleepiness	Epworth Sleepiness Scale (ESS) [12]	8 questions 4-point scale Global score ranging from 0 to 24	Recent times (past few weeks to few months)	↑ scores = ↑ level of daytime sleepiness 0–5 lower normal daytime sleepiness 6–10 higher normal daytime sleepiness 11–12 mild excessive daytime sleepiness 13–15 moderate excessive daytime sleepiness 16–24 severe excessive daytime sleepiness		
	Objective sleep parameters	Actiwatch [36] *Total sleep time, awakenings after sleep onset, sleep onset latency, sleep efficiency*				Sleep efficiency < 85%	sleep disturbance
PAIN	Pain intensity	McGill Pain Questionnaire short form (MPQ-SF) [13,40,48,54]	3 subscales Sensory and affective pain ratings on 4-point rating scale Pain intensity	Current Past month [54]	↑ scores = ↑ pain		
		Numeric (Pain) Rating Scale (NPRS) [13,42,49,50,52,56]	Single item	Past week	↑ scores = ↑ pain intensity Scale from 0 (no pain) to 10 (pain as bad as it could be/worst imaginable pain)		
		Visual Analogue Scale (VAS) [12,33,41,43,47]	Single 11-point item	Current Past week	↑ scores = ↑ pain intensity 0–4 no pain 5–44 mild pain 45–74 moderate pain 75–100 severe pain		
		SF-36 Bodily Pain Scale (BPS) [54]		Past 4 weeks	↑ scores = ↑ bodily pain		
		Self-reported questionnaire [48] *How intense was your pain?*	9-point scale	Past 3 h	0—not at all 2—somewhat 4—much 6—very much 8—extremely		
	Pain duration	Self-reported questionnaire [13,42,56] *Duration of pain*			<1 year ≥1 year		

**Table 9 jcm-10-03836-t009:** Brief overview of the main results of the cross-sectional data.

Pain	Sleep	Association	Strength of Conclusion	
			CLBP	CNP
Pain intensity	Sleep quality	−	** [40,42,43,45,52]	* [47]
	Insomnia severity	+	** [12,13,42,49,55,56]	* [50]
	Sleep disturbance Sleep duration Sleepiness Objective TST and SE Objective SOL and WASO	+	* [44]	* [11]
		−	* [11]	
		+	* [55]	
		−	* [42]	
		+	* [42]	
Pain duration	Sleep quality Insomnia severity Insomnia severity Sleepiness	No	** [41,43,54]	* [50]
		No	* [12,49]	
		−	* [55]	
		−	* [55]	
↑ odds of CLBP versus controls	Insomnia severity Sleep duration Sleep sufficiency	+	* [46]	
		−	* [51,57]	
		−	* [57]	
CLBP	↑ odds of poor sleep quality versus good sleep quality ↑ odds of sleep disturbances versus no sleep disturbances	−	* [43]	

		+	* [41]	
CNP (whiplash)	↑ odds of sleep problems versus no sleep problems	+		* [39]

↑, higher; CLBP, chronic low back pain; CNP, chronic neck pain; SE, sleep efficiency; SOL, sleep onset latency; TST, total sleep time; WASO, wake after sleep onset; −, negative association; +, positive association, No, no association; * weak evidence; ** moderate evidence.

**Table 10 jcm-10-03836-t010:** Brief overview of the main results of the longitudinal data.

Independent Variable	Dependent Variable	Association	Strength of Conclusion	
			CLBP	CNP
Sleep problems	Occurrence of CSP	+	** [33,35,38]	** [34,35,38]
	Pain intensity at FU	+	* [36]	
	Odds of non-recovery	+	* [36]	
Insomnia symptoms	Risk of non-recovery	+	* [37]	
CLBP	Insomnia	No	* [33]	

↑, higher; CLBP, chronic low back pain; CNP, chronic neck pain; CSP, chronic spinal pain; FU, follow-up; +, positive association, No, no association; * weak evidence; ** moderate evidence.

## Data Availability

Not applicable.

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
