# Peer review of "The Association between Sleep and Chronic Spinal Pain: A Systematic Review from the Last Decade"

_jcm, 2021, doi:10.3390/jcm10173836_

Round 1

Reviewer 1 Report

The authors did a good job addressing comments provided previously by this reviewer.

Further review brought up some additional comments for the authors:

It would be helpful for the reader to have a visual or table representation of the relationships identified in the results section.

The authors may want to discuss the potentially confounding relationship of increased pain in the morning as related to timing of pain medication and how limits on this data may impact interpretation.

Author Response

Dear reviewer,

Thank you again for taking the time to review our manuscript. Following your suggestions, a few more changes were made to the manuscript

Remark 1: It would be helpful for the reader to have a visual or table representation of the relationships identified in the results section.

Respone 1: Thank you for this relevant suggestion. We agree that adding a clear figure or table will make it easier for the reader to interpret the multiple results. Therefore, Table 9 for the cross-sectional findings and Table 10 for the longitudinal results were added after the corresponding sections, with refernece to it under the various subheadings of the results summaries (subheadings 3.5.5, 3.6.1 and 3.7.1). 

Remark 2: The authors may want to discuss the potentially confounding relationship of increased pain in the morning as related to timing of pain medication and how limits on this data may impact interpretation.

Response 2: The reviewer made a critical comment that the interpretation of the results regarding the course of pain during the day may be strongly influenced by the intake of pain medication. This point is addressed in the manuscript by the addition of a few sentences.

"In addition, the timing of possible pain medication intake may have influenced the findings of  more severe pain in the morning compared to the afternoon or evening [40, 48]. Thus, due to a lack of information on medication intake, these results are open to interpretation, as subjects may have taken analgesics after the morning assessment."

Ref 40: Heffner, K.; France, C.; Trost, Z.; Mei, H.; Pigeon, W. Chronic Low Back Pain, Sleep Disturbance, and Interleukin-6. Clin. J. Pain 2011, 2723, 35–41, doi:10.1038/jid.2014.371

Please find attachted the adapted manuscript.

Reviewer 2 Report

Looveren et al. have addressed all my concerns. In my opinion, the paper may be published in the current form.

Author Response

Dear reviewer,

Thank you again for taking the time to review our manuscript.

In response to the other reviewer's suggestions, a few additional adjustments were made to the manuscript.

Remark 1: It would be helpful for the reader to have a visual or table representation of the relationships identified in the results section.

Respone 1: Thank you for this relevant suggestion. We agree that adding a clear figure or table will make it easier for the reader to interpret the multiple results. Therefore, Table 9 for the cross-sectional findings and Table 10 for the longitudinal results were added after the corresponding sections, with refernece to it under the various subheadings of the results summaries (subheadings 3.5.5, 3.6.1 and 3.7.1).

Remark 2: The authors may want to discuss the potentially confounding relationship of increased pain in the morning as related to timing of pain medication and how limits on this data may impact interpretation.

Response 2: The reviewer made a critical comment that the interpretation of the results regarding the course of pain during the day may be strongly influenced by the intake of pain medication. This point is addressed in the manuscript by the addition of a few sentences.

"In addition, the timing of possible pain medication intake may have influenced the findings of more severe pain in the morning compared to the afternoon or evening [40, 48]. Thus, due to a lack of information on medication intake, these results are open to interpretation, as subjects may have taken analgesics after the morning assessment."

Ref 40: Heffner, K.; France, C.; Trost, Z.; Mei, H.; Pigeon, W. Chronic Low Back Pain, Sleep Disturbance, and Interleukin-6. Clin. J. Pain 20112723, 35–41, doi:10.1038/jid.2014.371

Please find attachted the adapted manuscript.

This manuscript is a resubmission of an earlier submission. The following is a list of the peer review reports and author responses from that submission.

Round 1

Reviewer 1 Report

Looveren et al. took the important topic of the association between sleep and chronic spinal pain. In my opinion, there are some elements, which should be changed:

Major:
- What with the Google Scholar database? Why do the authors decide to avoid it?
- The lower back pain may be an effect of obesity. This group of patients use some drugs which may affect the sleep quality - in my opinion, it should be better discussed in the paper (e.g. 10.1016/j.smrv.2020.101380)
- The authors should provide a separate section pertaining to the strengths and limitations of the study.

Minor:
- There is no need to put the subsection titles to abstract.
- Table 2 may be presented in the supplementary materials, which are available for all MDPI authors.

Author Response

Response to Reviewer 1 Comments

Thank you for reviewing our manuscript. Listed below are the changes made to the manuscript in response to your comments. 

Major

Point 1: What with the Google Scholar database? Why do the authors decide to avoid it?

Response 1: Although Google Scholar is a powerful search system for finding specific literature, it is not regarded as suitable as principal search engine when conducting a systematic review. Therefore, we included four databases (i.e., Pubmed, Embase, Web of Science and PsycARTICLES) that are considered more relevant. Hereby, PsycARTICLES is more focused on behavioral and social science research, which is in line with the topic of this review. By using these databases, which primarly focus on peer-reviewed articles, there may be some publication bias as gray literature was not accounted for. The latter shortcoming has been added to the limiations section (see also Point 3).

Reference:

Gusenbauer, M.; Haddaway, N.R. Which academic search systems are suitable for systematic reviews or meta-analyses? Evaluating retrieval qualities of Google Scholar, PubMed, and 26 other resources. Res. Synth. Methods 2020, 11, 181–217, doi:10.1002/jrsm.1378.

Point 2: The lower back pain may be an effect of obesity. This group of patients use some drugs which may affect the sleep quality - in my opinion, it should be better discussed in the paper (e.g. 10.1016/j.smrv.2020.101380)

Response 2: 

These comments were taken into account in the section on confounders in the discussion. 

"Checking for age and sex, as well as for psychological factors such as anxiety and depression as confounders was incorporated into the screening for risk of bias. However, parameters such as medication use and comorbidities like obesity may also have impacted the results and should therefore be taken into account. Regarding this first parameter, in fact, not only sleep medications affect sleep quality. Several other types of drugs such as analgesics (e.g., opioids), cardiovascular medications, antidepressants, corticosteroids and statins can interfere with sleep [63,73,74]. In clinical practice, pharmacological approaches should therefore be adjusted in dosing and timing to minimize sleep disturbances [63]. Furthermore, overweight and obesity, frequently reported by body mass index (BMI), are assumed to be risk factors for low back pain [75]. Mechanisms underlying this association can be attributed to increased mechanical load on the spine, structural degenerations of the disc and vertebral endplate, and chronic inflammation [75]."

References:

63: Haack, M.; Simpson, N.; Sethna, N.; Kaur, S.; Mullington, J. Sleep deficiency and chronic pain: potential underlying mechanisms and clinical implications. Neuropsychopharmacology 2020, 45, 205–216, doi:10.1038/s41386-019-0439-z.

73: Szmyd, B.; Rogut, M.; BiaÅ‚asiewicz, P.; Gabryelska, A. The impact of glucocorticoids and statins on sleep quality. Sleep Med. Rev. 2021, 55, doi:10.1016/j.smrv.2020.101380.

74: Van Gastel, A. Drug-Induced Insomnia and Excessive Sleepiness. Sleep Med. Clin. 2018, 13, 147–159, doi:10.1016/j.jsmc.2018.02.001.

75: Shiri, R.; Karppinen, J.; Leino-Arjas, P.; Solovieva, S.; Viikari-Juntura, E. The association between obesity and low back pain: A meta-analysis. Am. J. Epidemiol. 2010, 171, 135–154, doi:10.1093/aje/kwp356.

Point 3: The authors should provide a separate section pertaining to the strengths and limitations of the study.

Response 3: A seperate section on the strengths and limitations of the study was added to the manuscript. In addition, a few strengths were included to be complete. In the limitations section, several changes were made to comply with point 1 and point 2 of the comments.

Adjustments made:

- "Several limitations and strengths of the included studies and the review itself need to be considered when interpreting the results from this review."

- Adjustments mentioned under point 2

- "By only including articles published after January 2009, potentially previously published relevant articles on CNP were not included in the review. Since only a limited number of studies on neck pain were found, extending the search strategy to articles published before January 2009 might have led to stronger findings. By not including gray literature, some publication bias may also have occurred. Additionally, the research question in this systematic review differs somewhat from the one of Kelly et al. (2011) by focusing more specifically on the association between sleep and spinal pain, including longitudinal cohort studies on baseline healthy individuals [16]. However, due to the fact that most of the recorded studies were cross-sectional, it remains difficult to draw firm conclusions on causality."

- "In addition to the limitations, some strengths should be mentioned. This review added patients with chronic neck pain to the study population in examining the association between sleep and pain, which had not been examined to date. Furthermore a systematic approach based on the PRISMA guidelines was applied. In adopting this approach, the different phases of conducting the review (i.e., literature screening, risk of bias scoring, collection of results) were performed by two independent researchers, thus limiting detection and performance bias."

Minor

Point 4: There is no need to put the subsection titles to abstract.

Response 4: The subsection titles were removed from the abstract. A sentence was also omitted to avoid repetition within the abstract.

The following sentences

"The majority of studies reported an association between sleep parameters and chronic spinal pain, linking more severe sleep problems to higher pain levels. Weak to moderate evidence is present for an association between sleep variables and chronic spinal pain, with more severe pain accompanied by more disturbed sleep."

were replaced by

"Weak to moderate evidence is present for an association between sleep variables and chronic spinal pain, with more severe pain accompanied by more disturbed sleep. "

Point 5: Table 2 may be presented in the supplementary materials, which are available for all MDPI authors.

Response 5: Table 2 was added to the supplementary material and can thus be found as Appendix 1. Following this, the numbering of the other tables were changed so that they are sequential.

Additional points

Some modifications were made to correct and complete references in Tables 2, 3, 4 and 7. Several spelling errors were also corrected.

Reviewer 2 Report

The article provides a review and summary of the past decade of research on the association between sleep and chronic spinal pain. The authors provide easy to read tables to summarize the articles reviewed. The methodology of the review is succinct and appropriately descriptive.  Overall the review is well written and needs only minor grammatical or syntax editing. In Table 3, 4, and 5 "too less information" should be written as "too little information." Table 6, "B"  this description should be "not with all characteristics."

Author Response

Response to Reviewer 2 Comments

Thank you for reviewing our manuscript. Listed below are the changes made to the manuscript in response to your comments. 

Point 1: In Table 3, 4, and 5 "too less information" should be written as "too little information."

Response 1: The grammatical error was adjusted in the legends of the tables mentioned.

Point 2: Table 6, "B"  this description should be "not with all characteristics."

Response 2: The word ‘al’ was modified to 'all' in Table 6.

Additional points

Some modifications were made to correct and complete references in Tables 2, 3, 4 and 7. Several spelling errors were also corrected.
